# ProteinAdapter: Adapting Pre-trained Large Protein Models for Efficient Protein Representation Learning

## Abstract

The study of proteins is crucial in various scientific disciplines, but understanding their intricate multi-level relationships remains challenging. Recent advancements in Large Protein Models (LPMs) have demonstrated their ability in sequence and structure understanding, suggesting the potential of directly using them for efficient protein representation learning. In this work, we introduce **ProteinAdapter**, to efficiently transfer the general reference from the multiple Large Protein Models (LPMs), *e.g.*, ESM-1b, to the task-specific knowledge. ProteinAdapter could largely save labor-intensive analysis on the 3D position and the amino acid order. We observe that such a simple yet effective approach works well on multiple downstream tasks. Specifically, (1) with limited extra parameters, ProteinAdapter enables multi-level protein representation learning by integrating both sequence and geometric structure embeddings from LPMs. (2) Based on the learned embedding, we further scale the proposed ProteinAdapter to multiple conventional protein tasks. Considering different task priors, we propose a unified multi-scale predictor to fully take advantage of the learned embeddings via task-specific focus. Extensive experiments on over 20 tasks show that ProteinAdapter outperforms state-of-the-art methods under both single-task and multi-task settings. We hope that the proposed method could accelerate the study of protein analysis in the future.

## 1 Introduction

Proteins serve as vital constituents and functional units for life, underscoring the significance of protein research for life sciences. As understanding their intricate structure-function relationships remains costly and time-consuming, there is an urgent need to develop a discriminative protein representation for enhancing various computational biological analyses.

Recently, Large Protein Models (LPMs) have been verified as superior protein representation learners from different structure levels, containing 1D Protein Language Models (PLMs) (Rives et al., 2021; Meier et al., 2021; Lin et al., 2022) and 3D Protein Structure Models (PSMs) (Hsu et al., 2022; Zhang et al., 2023c). These works motivate us to ride on the coattails of LPMs, seize the windfalls of multi-level proteomic knowledge, and explore multi-level protein embeddings for efficient protein representation learning. However, considering the complex characteristics of proteins, there remain two key challenges: (1) *multi-level complementarity*, and (2) *multi-scale integration*. **On the one hand**, for various protein tasks, different levels of structure, such as amino acid sequences (primary structure) and geometric coordinates (tertiary structure) exhibit complementarity. Namely, for the protein-protein interaction prediction task, 1D sequence information can be used to predict potential protein partners, while 3D coordinate information can further help clarify the specific details of these interactions. **On the other hand**, a protein can encompass multiple smaller substructures that are meaningful at different scales. Namely, for the protein function classification task, the sequence lengths of different functional regions or domains exhibit variability, ranging from a few amino acid residues to hundreds.

In an attempt to address both limitations, we propose a new multi-level adapter, called **ProteinAdapter**, to take advantage of existing pre-trained large models for efficient protein representation learning. Our ProteinAdapter explicitly captures the interrelations and complementarity among multi-

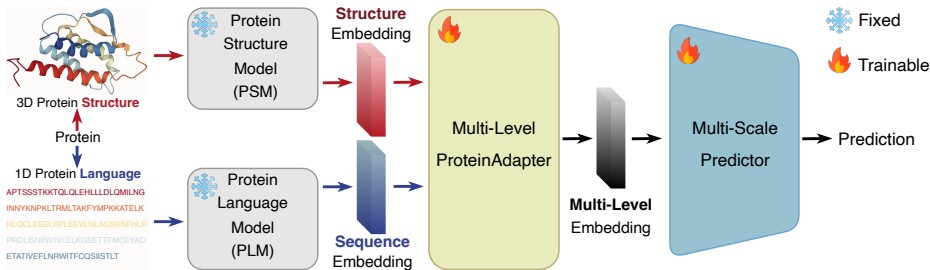

Figure 1: Overview of the proposed framework. The primary 1D sequence and the tertiary 3D structure are first individually fed into two Large Protein Models (LPMs) to obtain their corresponding embeddings. The proposed ProteinAdapter is to fuse the complementary embeddings from both levels to obtain the multi-level embedding. Finally, the merged multi-level embedding is fed into a multi-scale predictor to further take both local and global protein property into consideration.

level protein representations. Specifically, (1) to facilitate *multi-level interrelations*, the proposed ProteinAdapter directly takes the intermediate feature from pretrained ESM-1b and ESM-IF1 as inputs, and outputs a multi-level mixed representation embedding containing knowledge from both 1D and 3D structure levels. ProteinAdapter consists of stacked Cross-Self Attention (CSA) layers, in which the cross-attention block effectively mixes the multi-level protein embeddings while the self-attention block aims to preserve the original local-global representation capabilities. (2) Aiming at *multi-scale integration*, as a minor contribution, we further design a multi-scale predictor for various downstream tasks. Specifically, the predictor adopts a hierarchical pyramid structure that dynamically adjusts the weights between different sizes to ensure a comprehensive understanding of the mixed representation. Furthermore, different from conventional multi-task methods using single-scale generic representations, our multi-scale predictor naturally enables applicability in multi-task scenarios. Our contributions can be summarized as follows:

- A multi-level adapter, dubbed **ProteinAdapter**, is proposed for parameter-efficient fine-tuning on pre-trained large protein models, which effectively merges protein embeddings from different structure levels.

- A multi-scale hierarchical predictor is further designed to utilize the merged multi-level protein representation. Compared to previous works, our method can efficiently avoid tedious pretraining, and effectively transfer the multi-level knowledge for various downstream tasks.

- Due to the use of frozen protein models and the lightweight adapter, our method is more compute-efficient than existing state-of-the-arts. Extensive experiments on over 20 tasks show that the proposed method surpasses previous methods by large margins. Furthermore, our ProteinAdapter trained in a multi-task setting still consistently outperforms existing counterparts on most tasks, indicating the ability of protein representation and generalization.

## 2 RELATED WORK

**Single-Level Protein Representation Learning.** Proteins exhibit multi-level structures. Existing protein representation methods mainly focus on the 1D primary and the 3D tertiary structures understanding. The primal is the protein sequence consisting of amino acids, and the tertiary describes the natural folded three-dimensional structure, For the primary structure, regarding protein sequences as the language of life, many Protein Language Models (PLMs) (Bepler & Berger, 2019; Strodthoff et al., 2020; Vig et al., 2020; Rives et al., 2021; Amidi et al., 2018; Elnaggar et al., 2023) have been proposed for sequence-based protein representation learning with large-scale protein sequence corpora. For the tertiary structure, several Protein Structure Models (PSMs) (Hsu et al., 2022; Fan et al., 2022; 2023; Hermosilla et al., 2020; Zhang et al., 2023c) propose to extract features directly from the geometric information of amino acids or atoms. In this work, we directly use the power of these well-trained Large Protein Models (LPMs) to acquire discriminative protein embeddings.

**Multi-level Protein Representation Learning.** Since each structure of the protein has its own merit and driving forces in describing specific characteristics, several works (Wang et al., 2023; Zhang et al., 2023b; Fan et al., 2022; 2023) have been proposed to explicitly model the complementary information between different levels. These methods are typically designed based on Graph Neural Networks (GNNs) (Scarselli et al., 2008) or Convolution Neural Networks (CNNs) (Krizhevsky et al., 2012). However, due to the computation limits, these methods usually process adjacent amino acids within limited neighboring graph nodes (Zhang et al., 2023c; Sun et al., 2022) or small convolution kernels (Fan et al., 2022; 2023), and deploy stacked downsampling to expand the receptive field for global perception. The former local propagation pattern with a fixed small region limits the perception field for protein functional regions. The latter downsampling operations with fixed length also break the relation between neighbor functional regions in proteins. Different from these methods, our ProteinAdapter leverages the trend of large models by directly utilizing two single-level models focused on different structures (*i.e.*, ESM-1b (Rives et al., 2021) for 1D sequence and ESM-IF1 (Hsu et al., 2022) for 3D coordinates), and achieves multi-level protein representations through an efficient adapter.

**Parameter Efficient Fine-Tuning (PEFT).** To utilize the rich evolutionary and biological patterns from these pretrained LPMs, ESM-GearNet (Zhang et al., 2023b) makes the first attempt by replacing the original graph node with the well-learned 1D sequence embedding produced by ESM-1b (Rives et al., 2021). However, this method still requires tedious training of a complex local structure encoder from scratch. To fully unleash and efficiently utilize the power of large models, Parameter-Efficient Fine-Tuning (PEFT) methods (Li & Liang, 2021; Lester et al., 2021; Liu et al., 2022; Houlsby et al., 2019; Hu et al., 2021) have prevailed recently in both Natural Language Processing (NLP) and Computer Vision (CV) communities. Existing PEFT methods can be roughly divided into three parts: Prefix-tuning (Li & Liang, 2021; Lester et al., 2021; Liu et al., 2022), Adapter-tuning (Houlsby et al., 2019), and LoRA (Hu et al., 2021). In this paper, we resort to the adapter by adding only a few trainable parameters for different downstream protein tasks, while the parameters of the original LPMs are fixed. To the best of our knowledge, this is the first adapter-based work exploring the parameter-efficient fine-tuning of pre-trained large protein models.

## 3 METHODS

Aiming at efficient utilization of off-the-shelf pre-trained models for protein representation learning, there are two key properties that differentiate protein language from natural language: *multi-level* and *multi-scale*. First, a protein can target various structural levels to carry out its functions. Each level possesses distinct advantages and underlying factors in elucidating particular attributes. Consequently, it is necessary to consider both 1D and 3D structures for comprehensive protein representation. Second, due to their distinct biological functions, the scale of functional regions in different proteins typically varies. Likewise, due to genetic differences, the scale of the same functional regions is often inconsistent across different species. Thus the predictor should possess multi-scale perceptual capabilities to fully leverage the protein representation for various downstream tasks. Considering these two properties, as shown in Fig. 1, our method consists of three key components: pre-trained protein models, a multi-level ProteinAdapter, and a multi-scale predictor.

### 3.1 ACQUIRING MULTI-LEVEL PROTEIN EMBEDDINGS WITH PRE-TRAINED MODELS

Recently, large Protein Language Models (PLMs) (Vig et al., 2020; Rao et al., 2020; Rives et al., 2021) have demonstrated strong capabilities in understanding protein sequences, which encourages us to leverage pre-trained sequence embeddings with rich information In our approach, we use a powerful PLM, ESM-1b (Rives et al., 2021), as our 1D protein encoder, which takes protein sequences as input and outputs the sequence embedding $E_{seq}$.

However, considering that PLMs do not directly incorporate protein structures as input, they are limited in capturing intricate structural features. Given the importance of protein structures in determining functions, we adopt another Protein Structure Model (PSM), ESM-IF1(Hsu et al., 2022), as the 3D structure encoder. As a multi-level complement to the sequence embedding, the structure embedding $E_{str}$ obtained with ESM-IF1 effectively encapsulates the geometric information on proteins within the sequence embedding $E_{seq}$.

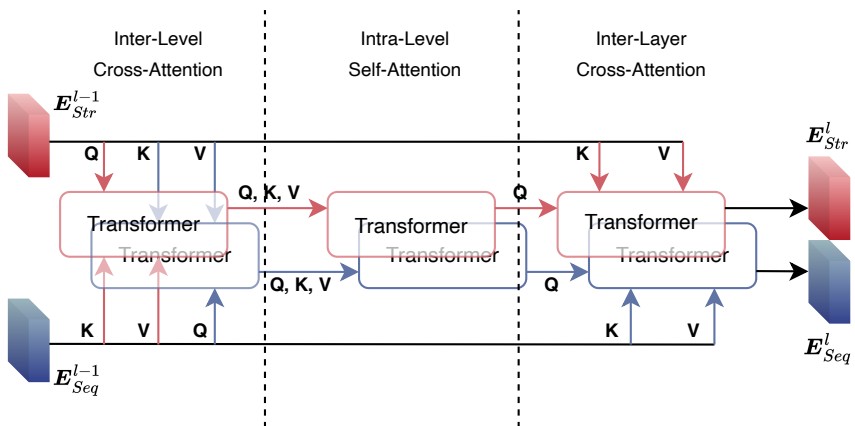

Figure 2: Architecture of the ProteinAdapter. In each layer, ProteinAdapter first utilizes an inter-level Cross-Attention block to inject interactive information from another level of embeddings. Then, each level separately undergoes the Intra-Level Self-Attention to improve its own embedding capability. Finally, the embedding of each level performs an Inter-Layer Cross-Attention with its corresponding embedding from the previous layer to avoid the forgetting issue between layers.

## 3.2 Integrating Sequence-Structure Features with ProteinAdapter

Despite the excellent performance of PLMs and PSMs in sequence and structure understanding, they are individually trained on single-level protein data. This implies the need to effectively integrate their embeddings while minimizing the reduction in their respective unimodal representation capabilities. To preserve such unimodal information when injecting the cross-level information from each other, in this subsection, we propose to integrate sequence-structure features with a new ProteinAdapter. As shown in Fig. 2, ProteinAdapter is composed of multiple stacked layers, each layer containing alternate Cross-Attention (CA) and Self-Attention (SA) blocks. To facilitate the description, we use the **structural branch** (red part) in Fig. 2 as an example in the following part. The sequence branch (blue part) follows the same pattern as the structural branch, with the only difference being the reversal of token selection in cross-attention blocks.

**Inter-Level Cross-Attention.** First, given the structure embedding $\boldsymbol{E}_{str}^{l-1}$ and sequence embedding $\boldsymbol{E}_{seq}^{l-1}$ from the previous layer, this block takes $\boldsymbol{E}_{str}^{l-1}$ as the input queries $\mathbf{Q}$ and $\boldsymbol{E}_{seq}^{l-1}$ as the keys $\mathbf{K}$ and values $\mathbf{V}$, and produces a refined structure embedding that contains sequence information. Specifically, $\mathbf{Q}$, $\mathbf{K}$, $\mathbf{V}$ are acquired with linear transformation using learned weight matrices as:

$$\mathbf{Q} = \boldsymbol{E}_{str}^{l-1}\mathbf{W}_q, \mathbf{K} = \boldsymbol{E}_{seq}^{l-1}\mathbf{W}_k, \mathbf{V} = \boldsymbol{E}_{seq}^{l-1}\mathbf{W}_v, \quad (1)$$

Where $\mathbf{W}_q$, $\mathbf{W}_k$, and $\mathbf{W}_v$ are the learned weight matrices for the $\mathbf{Q}$, $\mathbf{K}$, $\mathbf{V}$, respectively. Next, the scaled attention scores $\alpha$ are computed by taking the dot product of the transformed queries $\mathbf{Q}$ from the structure embedding and keys $\mathbf{K}$ from the sequence embedding as:

$$\alpha = \frac{\mathbf{Q}\mathbf{K}^T}{\sqrt{d_k}}, \quad (2)$$

where $d_k$ represents the dimensionality of the keys. Using the attention scores, we compute the attention weights $w$ via softmax and then derive the output $\mathbf{X}_{inter}$ using a weighted sum of the transformed values $\mathbf{V}$. It can be formulated as:

$$\mathbf{X}_{inter} = Softmax(\alpha) \times \mathbf{V}. \quad (3)$$

In practice, due to the differing dimensions of the initial multi-level embedding ($\boldsymbol{E}_{str} \in \mathbb{R}^{N \times 512}$, $\boldsymbol{E}_{seq} \in \mathbb{R}^{N \times 768}$), we adjust the $\mathbf{K}$, $\mathbf{V}$ to the same dimension with $\mathbf{Q}$ using the linear transformation weights (*i.e.*, $\mathbf{W}_k$, $\mathbf{W}_v \in \mathbb{R}^{768 \times 512}$) in the first adapter layer.

**Intra-Level Self-Attention.** Next, an Intra-Level Self-Attention block is employed to ensure the quality of the original embeddings. This block follows similar operations on $\mathbf{Q}$, $\mathbf{K}$ and $\mathbf{V}$, except that they all come from the output $\mathbf{X}_{intra}$ of the previous cross-attention block, which is formulated as:

$$\mathbf{X}_{intra} = Softmax\left(\frac{\mathbf{X}_{inter}\mathbf{W}_q(\mathbf{X}_{inter}\mathbf{W}_k)^T}{\sqrt{d_k}}\right)\mathbf{X}_{inter}\mathbf{W}_v \quad (4)$$

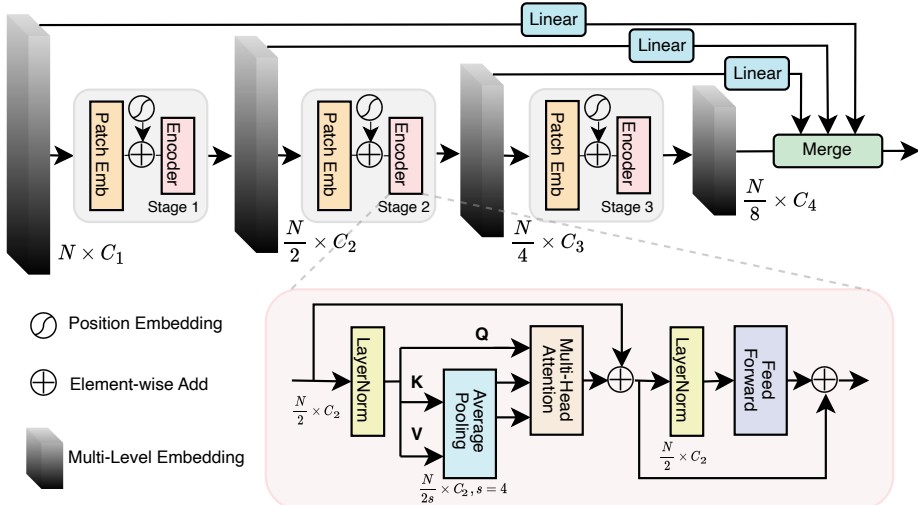

Figure 3: Architecture of the Multi-Scale Predictor. In each stage, the input embedding first undergoes an overlapping patch embedding block for $2\times$ downsamping, and then undergoes the encoder for multi-head self-attention. We deploy an efficient transformer as the Encoder structure, where the key and value is downsampled with $s = 4$.

**Inter-Layer Cross-Attention.** Then, an Inter-Layer Cross-Attention block is further deployed to avoid the forgetting problem across multiple layers. As can be seen from Fig. Fig. 2, this block explicitly considers the information from the previous structure embedding $\boldsymbol{E}_{str}^{l-1}$ as $\mathbf{K}$ and $\mathbf{V}$, and further refines the output $\boldsymbol{E}_{str}^{l}$ to the next layer. It can be formulated as:

$$\boldsymbol{E}_{str}^{l} = Softmax\left(\frac{\mathbf{X}_{intra}\mathbf{W}_q(\boldsymbol{E}_{str}^{l-1}\mathbf{W}_k)^T}{\sqrt{d_k}}\right)\boldsymbol{E}_{str}^{l-1}\mathbf{W}_v \tag{5}$$

Finally, considering amino acids primarily react with their surrounding neighbors, we further employ 1D convolution with sliding windows after the last adapter layer, to extract local features from the concatenated multi-level embeddings. After such alignment, the PSM is enriched by the valuable property information within PLM, to capture the protein property information with different levels while preserving its original representation powers.

### 3.3 ENHANCING PYRAMIDAL REPRESENTATIONS WITH MULTI-SCALE PREDICTOR

After thoroughly fusing the sequence and geometric embeddings through ProteinAdapter, the obtained multi-level embedding contains rich evolutionary and biological patterns underlying both levels of protein structures. Now we move a further step to fully utilize the multi-level embedding for downstream tasks. As shown in Fig. 3, we design a multi-scale predictor to efficiently extract and utilize protein features in a hierarchical manner.

**Patch Embedding.** During each stage, an overlapping patch embedding block (Wang et al., 2021a; 2022) with downsampling rate 2 is performed to halve the protein resolution and build a pyramid architecture. Specifically, given the embedding $\boldsymbol{E}_{i-1} \in \mathbb{R}^{N_{i-1} \times C_{i-1}}$ from the previous stage $i-1$ as input, we feed it to a 1D convolution to acquire the downsampled embedding $\boldsymbol{E}_i \in \mathbb{R}^{N_i \times C_i}$, where $N_i$ is computed as $N/2^i$. In our implementation, we set channel numbers $C_1, C_2, C_3, C_4$ as 512, 512, 1024, 1024. The 1D convolution is set as the stride 2, kernel size 3, zero padding size of 1, and the kernel number of $C_i$. This downsampling patch embedding block flexibly adjusts the feature scale in each stage, making it possible to construct a feature pyramid for the Transformer. In this way, our method can handle longer input protein embedding with limited resources.

**Transformer Encoder.** Following existing efficient transformer methods (Wang et al., 2022; Han et al., 2023), we adopt zero padding position encoding into the transformer encoder. In our implementation, we further use a linear attention mechanism (Katharopoulos et al., 2020) that uses average pooling to reduce the sequence length $n$ with a pooling size $s = 4$ before the attention operation, as shown in Fig. 3. We also employ a focused function (Han et al., 2023)on the attention map, to

pay more attention to those effective functional regions. In our implementation, we use dynamic batching (Rives et al., 2021) to handle variable-length sequences for higher computational efficiency.

**Multi-Scale Prediction.** Finally, the features from various scales are linearly mapped to the dimensions corresponding to downstream tasks. Then, through a set of learnable parameters, the weights of different scales are dynamically adjusted, thereby achieving adaptability to different tasks and the possibility of multi-task learning.

## 4 EXPERIMENTS

### 4.1 EVALUATION TASKS AND DATASETS

To validate the efficacy of ProteinAdapter, we conducted tests on over 20 tasks. Since our model explicitly leverages multi-level protein data, we separately assessed the model's ability to protein sequences and structures understanding on the PEER (Xu et al., 2022) and ATOM3D (Townshend et al., 2020) benchmarks. Subsequently, we evaluated the performance in a multi-task setting on the PEER benchmark, verifying its generalization capability. We then compared the efficiency superiority of ProteinAdapter against the current LPMs. Finally, we carried out ablation studies on four sequence-structure paired tasks, confirming the effectiveness of the proposed multi-level adapter and multi-scale predictor.

PEER (Xu et al., 2022) is a comprehensive and multi-task benchmark for protein sequence understanding. ATOM3D (Townshend et al., 2020) is a structure-based benchmark designed primarily for 3D structural prediction and related tasks in molecular biology. The benchmark encompasses a wide array of 3D structures of proteins, small molecules, and protein-ligand complexes. We select the 22 kinds of the most common tasks from these two benchmarks, focusing on different tasks including localization prediction, function prediction, structure prediction, Protein-Protein Interaction (PPI) prediction, and Protein-Ligand Interaction (PLI) prediction, as shown in Table 1. More details on each task and metric can be found in Appendix A and Appendix B.

### 4.2 TRAINING SETUP

For the sequence-based PEER benchmark, due to the lack of corresponding 3D coordinate information in the datasets of the PEER benchmark, we simply replaced the structural embedding with another layer of sequence embedding (*i.e.*, layer 32 and 33 of ESM-1b). Another alternative approach is to use a pre-trained structural prediction model ESMFold (Lin et al., 2022) to generate structures. Due to computational limitations, we only conducted multi-level performance tests on the Stability Prediction and Fold Classification tasks. The Stability Prediction dataset has the shortest average sequence length (less than 70). For the Fold Classification task, we can collect the corresponding structures from RCSB-PDB (Berman et al., 2000b). Moreover, for the PPI task that takes paired proteins as inputs, we take the multi-level embeddings obtained for each of the two proteins, and then feed them into an MLP predictor defined based on the concatenation of the embeddings of the two proteins. For the PLI task which has protein-ligand inputs, we follow previous practices (Hu et al., 2019; Sun et al., 2019), and involve an additional Graph Isomorphism Network (GIN) (Xu et al., 2018) with 4 layers and 256 hidden dimensions as the ligand graph encoder. As for the ATOM3D benchmark (Townshend et al., 2020), we can directly extract their protein sequences from the mdb or pdb files. Our method is implemented based on PyTorch 1.13.1 with CUDA 11.7. All experiments are conducted on two NVIDIA Tesla-V100 (32GB) GPU. As for other compared methods, we use the same default training settings in the benchmark.

### 4.3 SINGLE-TASK *vs.* MULTI-TASK

**Single-Task Learning.** Given a task $t \in \mathcal{T}$ from the pool $\mathcal{T}$ of benchmark tasks, a task-specific loss function $\mathcal{L}_t$ is defined to measure the correctness of model predictions on training samples against ground truth labels. The objective of learning this single task is to optimize model parameters to minimize the loss $\mathcal{L}_t$ on this task.

**Multi-Task Learning.** We further delve deeper into multi-task learning to validate the generalization ability of ProteinAdapter. Similar to PEER(Xu et al., 2022), our training objective comprises a

Table 1: Task descriptions. Each task, along with its acronym, category, the size of each dataset split, and evaluation metric are shown below. *Abbr.*, Reg.: regression; Cls.: classification; $R_S$: Spearman correlation; $R_P$: Pearson correlation; Acc: accuracy; RMSE: root-mean-square error; AUROC: area under the receiver operating characteristic curve; MAE: mean absolute error.

| Name (Acronym) | Task Category | Dataset Split | Metric |
|---|---|---|---|
| **PEER Benchmark** | | | |
| Fluorescence prediction (Flu) | Protein-wise Reg. | 21,446 / 5,362 / 27,217 | $R_S$ |
| Stability prediction (Sta) | Protein-wise Reg. | 53,571 / 2,512 / 12,851 | $R_S$ |
| $\beta$-lactamase activity prediction ($\beta$-lac) | Protein-wise Reg. | 4,158 / 520 / 520 | $R_S$ |
| Contact prediction (Cont) | Residue-pair Cls. | 25,299 / 224 / 40 | L/5 precision |
| Subcellular localization prediction (Sub) | Protein-wise Cls. | 8,945 / 2,248 / 2,768 | Acc |
| Binary localization prediction (Bin) | Protein-wise Cls. | 5,161 / 1,727 / 1,746 | Acc |
| Solubility prediction (Sol) | Protein-wise Cls. | 62,478 / 6,942 / 1,999 | Acc |
| Fold classification (Fold) | Protein-wise Cls. | 12,312 / 736 / 718 | Acc |
| Secondary structure prediction (SSP) | Residue-wise Cls. | 8,678 / 2,170 / 513 | Acc |
| Yeast PPI prediction (Yst) | Protein-pair Cls. | 1,668 / 131 / 373 | Acc |
| Human PPI prediction (Hum) | Protein-pair Cls. | 6,844 / 277 / 227 | Acc |
| PPI affinity prediction (Aff) | Protein-pair Reg. | 2,127 / 212 / 343 | RMSE |
| PLI prediction on PDBbind (PDB) | Protein-ligand Reg. | 16,436 / 937 / 285 | RMSE |
| PLI prediction on BindingDB (BDB) | Protein-ligand Reg. | 7,900 / 878 / 5,230 | RMSE |
| **ATOM3D Benchmark** | | | |
| Small Molecule Properties (SMP) | Protein-wise Reg. | 103,547 / 12,943 / 12,943 | MAE |
| Protein Interface Prediction (PIP) | Protein-pair Cls. | 1,240 / 155 / 155 | AUROC |
| Residue Identity (RES) | Protein-wise Cls. | 21,147 / 964 / 3,319 | Acc |
| Mutation Stability Prediction (MSP) | Protein-wise Cls. | 1,660 / 210 / 210 | AUROC |
| Ligand Binding Affinity (LBA) | Protein-ligand Reg. | 3,678 / 460 / 460 | RMSE |
| Ligand Efficacy Prediction (LEP) | Protein-ligand Cls. | 4,220 / 501 / 870 | AUROC |
| Protein Structure Ranking (PSR) | Protein-wise Reg. | 508 / 85 / 56 | $R_S$ |
| RNA Structure Ranking (RSR) | Protein-wise Reg. | 13,182 / 4,056 / 4,056 | $R_S$ |

*primary task* and a *supportive task*. Following the principle that "protein structures determine their functions" (Hegyi & Gerstein, 1999), we employ three structure prediction tasks, *i.e.*, contact prediction, fold classification and secondary structure prediction, as the auxiliary task. In more detail, when presented with a primary task $t_p$ characterized by loss $\mathcal{L}t_p$ and a supportive task $t_s$ characterized by loss $\mathcal{L}t_s$, our multi-task model follows the framework of hard parameter sharing (Ruder, 2017). Within this framework, we utilize a universal protein sequence encoder for all tasks, and a ligand graph encoder for protein-ligand interaction prediction tasks. Throughout the multi-task learning process, the network parameters are fine-tuned based on the combined loss of the primary and supportive tasks: $\mathcal{L} = \mathcal{L}t_p + \alpha\mathcal{L}t_s$. Here, $\alpha$ represents the balancing factor for the two objectives, defaulting to 1.0 unless otherwise mentioned. The training iterations are consistent with single-task learning on the primary task, and we only evaluate one primary task to maintain consistency with the PEER benchmark.

**Optimization Objective.** Following PEER Benchmark (Xu et al., 2022), the fluorescence, stability, $\beta$-lactamase activity, PPI affinity, PDBbind and BindingDB prediction tasks are trained with Mean Squared Error (MSE); the solubility, subcellular localization, binary localization, fold, secondary structure, yeast PPI and human PPI prediction tasks are trained with cross-entropy loss; the contact prediction task is trained with binary cross-entropy loss. As for ATOM3D benchmark (Townshend et al., 2020), we use MSE loss for regression tasks and cross-entropy loss for classification tasks.

## 4.4 COMPARISONS

**Evaluation on Single-Task Learning.** Table 2 and Table 3 respectively show the performance of ProteinAdapter on the PEER (Xu et al., 2022) and ATOM3D (Townshend et al., 2020) benchmarks at different levels. It can be observed that for sequence-based tasks in the PEER benchmark, simply interacting with sequence embeddings from different layers through our adapter can achieve competitive performance. With the addition of structural information, our performance is further

Table 2: PEER benchmark results on single-task learning. "-" indicates a non-applicable setting. The best performance is marked in **blod** and the second performance is underlined. ↑ indicates that a higher value corresponds to better performance.

| Task | Sequence Encoder | | Pre-trained LPMs | | Literature | ProteinAdapter |
|------|------------|-----|---------|--------|------------|----------------|
| | Transformer | CNN | ProtBert | ESM-1b | SOTA | (with structure embedding) |
| **Function Prediction** | | | | | | |
| Flu ↑ | 0.643 | 0.682 | 0.679 | 0.678 | 0.690, Shallow CNN (Shanehsazzadeh et al., 2020) | **0.698** |
| Sta ↑ | 0.649 | 0.639 | 0.771 | 0.694 | 0.790, Evoformer (Hu et al., 2022) | 0.782 (**0.805**) |
| $\beta$-lac ↑ | 0.261 | 0.781 | 0.731 | 0.839 | **0.890**, ESM-1b (Rives et al., 2021) | 0.887 |
| Sol ↑ | 70.12 | 64.43 | 68.15 | 70.23 | 77.00, DeepSol (Khurana et al., 2018) | **77.58** |
| **Localization Prediction** | | | | | | |
| Sub ↑ | 56.02 | 58.73 | 76.53 | 78.13 | **86.0**, LA-ProtT5 (Stärk et al., 2021) | 85.20 |
| Bin ↑ | 75.74 | 82.67 | 91.32 | 92.40 | 92.34, DeepLoc (Almagro Armenteros et al., 2017) | **93.55** |
| **Structure Prediction** | | | | | | |
| Cont ↑ | 17.50 | 10.00 | 39.66 | 45.78 | **82.10**, MSA Transformer (Rao et al., 2021) | 55.15 |
| Fold ↑ | 8.52 | 10.93 | 16.94 | 28.17 | 56.50, GearNet-Edge (Zhang et al., 2023c)) | 32.60 (**58.80**) |
| SSP ↑ | 59.62 | 66.07 | 82.18 | 82.73 | **86.41**, DML_SS (Yang et al., 2022) | 83.11 |
| **Protein-Protein Interaction Prediction** | | | | | | |
| Yst ↑ | 54.12 | 55.07 | 63.72 | 57.00 | — | **68.45** |
| Hum ↑ | 59.58 | 62.60 | 77.21 | 78.17 | — | **79.12** |
| Aff ↓ | 2.499 | 2.796 | 2.195 | 2.281 | — | **2.073** |
| **Protein-Ligand Interaction Prediction** | | | | | | |
| PDB ↓ | 1.455 | 1.376 | 1.562 | 1.559 | 1.181, SS-GNN (Zhang et al., 2023a) | **1.153** |
| BDB ↓ | 1.566 | 1.497 | 1.549 | 1.556 | **1.340**, DeepAffinity (Karimi et al., 2019) | 1.344 |

Table 3: ATOM3D benchmark results on single-task learning. "-" indicates a non-applicable setting. "*" indicates that the original implementation in ATOM3D benchmark. The best performance is marked in **blod** and the second performance is underlined. ↑ indicates that a higher value corresponds to better performance.

| Task | 3D | | | Non-3D | ProteinAdapter |
|------|--------|------|------|--------|----------------|
| | 3DCNN* | GNN* | ENN* | Method | |
| SMP ↓ | 0.754 | 0.501 | 0.052 | 0.496 (Tsubaki et al., 2019) | **0.021** |
| PIP ↑ | 0.844 | 0.669 | — | 0.841 (Sanchez-Garcia et al., 2019) | **0.875** |
| RES ↑ | 0.451 | 0.082 | 0.072 | 0.300 (Rao et al., 2019) | **0.531** |
| MSP ↑ | 0.574 | 0.609 | 0.574 | 0.554 (Rao et al., 2019) | **0.637** |
| LBA ↓ | 1.416 | 1.601 | 1.568 | 1.565 (Öztürk et al., 2018) | **1.057** |
| LEP ↑ | 0.589 | 0.681 | 0.663 | 0.696 (Öztürk et al., 2018) | **0.731** |
| PSR ↑ | 0.789 | 0.750 | — | 0.796 (Pagès et al., 2019) | **0.811** |
| RSR ↑ | 0.372 | 0.512 | — | 0.304 (Watkins et al., 2020) | **0.557** |

improved. As for the structure-based tasks in the ATOM3D benchmark, ProteinAdapter outperforms existing methods by a large margin, since our method explicitly utilizes the multi-level protein knowledge.

**Evaluation on Multi-Task Learning.** We then conducted multi-task learning experiments following the settings of the PEER benchmark, the results are shown in Table 4. It can be observed that since our method has already made full use of the structural information, using structure prediction tasks as an auxiliary does not significantly improve the performance of the central task. However, the results are sufficient to demonstrate that our method has decent generalization capability.

## 4.5  DISCUSSION

**Ablation Study.** In order to fully validate the effectiveness of our proposed multi-level adapter and multi-scale predictor, we conducted ablation experiments on two paired protein sequence-structure tasks: Gene Ontology Term Prediction and Enzyme Commission Number Prediction. We constructed two variants: (1) *w/o multi-level adapter*: remove the adapter and directly input the embeddings of the two LPMs into the predictor after concatenation; (2) *w/o multi-scale predictor*: remove the

Table 4: PEER benchmark results on multi-task learning. Red results outperform the original single-task learning baseline; gray results are the same as the baseline; blue results underperform the baseline; "-" indicates not applicable for this setting. Abbr., Ori.: original; Avg.: average performance under three auxiliary tasks. ↑ indicates that a higher value corresponds to better performance. The best performance among three auxiliary tasks is marked in **blod**.

| Task | Transformer | | | | | ESM-1b | | | | | ProteinAdapter | | | | |
|---|---|---|---|---|---|---|---|---|---|---|---|---|---|---|---|
| | Ori. | +Cont | +Fold | +SSP | Avg. | ori. | +Cont | +Fold | +SSP | Avg. | Ori. | +Cont | +Fold | +SSP | Avg. |
| **Function Prediction** | | | | | | | | | | | | | | | |
| Flu ↑ | 0.643 | 0.612 | 0.648 | **0.656** | 0.638 | 0.678 | **0.681** | 0.679 | **0.681** | 0.680 | 0.698 | 0.697 | 0.698 | **0.699** | 0.698 |
| Sta ↑ | 0.649 | 0.620 | **0.672** | 0.667 | 0.653 | 0.694 | 0.733 | 0.728 | **0.759** | 0.740 | 0.782 | 0.781 | **0.788** | 0.787 | 0.785 |
| $\beta$-lac ↑ | 0.261 | 0.142 | **0.276** | 0.197 | 0.205 | 0.839 | **0.899** | 0.882 | 0.881 | 0.887 | 0.887 | **0.897** | 0.891 | 0.889 | 0.892 |
| Sol ↑ | 70.12 | **70.03** | 68.85 | 69.81 | 69.56 | 70.23 | **70.46** | 64.80 | 70.03 | 68.43 | 77.58 | 77.57 | **77.60** | 77.51 | 77.56 |
| **Localization Prediction** | | | | | | | | | | | | | | | |
| Sub ↑ | 56.02 | 52.92 | **56.74** | 56.70 | 55.45 | 78.13 | **78.86** | 78.43 | 78.00 | 78.43 | 85.20 | **85.70** | 85.65 | 85.33 | 86.16 |
| Bin ↑ | 75.74 | 74.98 | **76.27** | 75.20 | 75.48 | 92.40 | **92.50** | 91.83 | 92.26 | 92.19 | 93.55 | 93.51 | **93.57** | 93.50 | 93.52 |
| **Structure Prediction** | | | | | | | | | | | | | | | |
| Cont ↑ | 17.50 | - | 2.04 | **12.76** | 7.40 | 45.78 | - | **35.86** | 32.03 | 33.94 | 55.15 | - | **55.21** | 55.17 | 55.19 |
| Fold ↑ | 8.52 | **9.16** | - | 8.14 | 8.65 | 28.17 | **32.10** | - | 28.63 | 30.36 | 32.60 | **33.21** | - | 33.18 | 33.19 |
| SSP ↑ | 59.62 | **63.10** | 50.93 | - | 57.0 | 82.73 | **83.21** | 82.27 | - | 82.74 | 83.11 | **83.15** | 83.10 | - | 83.13 |
| **Protein-Protein Interaction Prediction** | | | | | | | | | | | | | | | |
| Yst ↑ | 54.12 | 52.86 | **54.00** | **54.00** | 53.62 | 57.00 | 58.50 | **64.76** | 62.06 | 61.77 | 68.45 | 68.41 | **68.44** | 68.37 | 68.40 |
| Hum ↑ | 59.58 | 60.76 | **67.33** | 54.80 | 60.96 | 78.17 | 81.66 | 80.28 | **83.00** | 81.64 | 79.12 | 79.05 | **79.11** | 79.01 | 79.06 |
| Aff ↓ | 2.499 | 2.733 | **2.524** | 2.651 | 2.636 | 2.281 | **1.893** | 2.002 | 2.031 | 1.975 | 2.073 | 2.077 | 2.085 | **2.071** | 2.077 |
| **Protein-Ligand Interaction Prediction** | | | | | | | | | | | | | | | |
| PDB ↓ | 1.455 | 1.574 | 1.531 | **1.387** | 1.497 | 1.559 | 1.458 | 1.435 | **1.419** | 1.437 | 1.153 | 1.155 | **1.150** | 1.151 | 1.152 |
| BDB ↓ | 1.566 | 1.490 | **1.464** | 1.519 | 1.491 | 1.556 | 1.490 | 1.511 | **1.482** | 1.494 | 1.344 | 1.339 | **1.331** | 1.340 | 1.336 |

Table 5: Ablation and efficiency study on Gene Ontology (GO) term prediction and Enzyme Commission (EC) number prediction tasks. ↑ indicates that a higher value corresponds to better performance.

| Methods & Variants | GO | | | EC ↑ |
|---|---|---|---|---|
| | BP ↑ | MF ↑ | CC ↑ | |
| CDConv (Fan et al., 2022) | 0.453 | 0.654 | 0.479 | 0.820 |
| ESM-1b (650M) (Rives et al., 2021) | 0.452 | 0.657 | 0.477 | 0.864 |
| GearNet-Edge (with pre-training) (Zhang et al., 2023c) | 0.490 | 0.654 | 0.488 | 0.874 |
| w/o adapter (13.5M) | 0.457 | 0.658 | 0.477 | 0.868 |
| w/o predictor (15.5M) | 0.472 | 0.670 | 0.481 | 0.871 |
| **ProteinAdapter (27.3M)** | **0.507** | **0.681** | **0.502** | **0.881** |

predictor and directly input the fused embedding from the adapter into a three-layer MLP. The $F_{max}$ accuracy is used as the evaluation metric for these two tasks. As shown in Table 5, it can be seen that both the multi-level adapter and multi-scale predictor are indispensable, especially for complex multi-label tasks.

**Efficiency.** By integrating two pre-trained LPMs with our lightweight ProteinAdapter, our method can achieve higher performance with fewer parameters compared to training a single-level large model from scratch. As shown in Table 5, with the default setting of two layers in the adapter and three downsampling times in the predictor, our model has $20\times$ fewer parameters than the existing methods (27.3M of ProteinAdapter vs 650M of ESM-1b).

## 5 CONCLUSION

The proposed ProteinAdapter offers a scalable solution for parameter-efficient fine-tuning on large pre-trained protein models. The multi-level adapter seamlessly merges protein embeddings, eliminating the need for resource-intensive pretraining. The method demonstrates superior compute efficiency through frozen models and a lightweight adapter. Extensive experiments across 22 tasks showcase its substantial performance gains over existing methods. In a multi-task setting, ProteinAdapter consistently outperforms counterparts, establishing its versatility and generalization capabilities.

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

# A  DETAILS ON EVALUATION TASKS

## A.1  PEER BENCHMARK

**Subcellular localization prediction (Sub).**

*Impact:* Determining the subcellular positioning of a protein can significantly enhance target pinpointing in drug development (Rajagopal & Simon, 2003). A tool that predicts subcellular localization swiftly and precisely can expedite this procedure. This endeavor aids in the creation of such an instrument.

*Target:* The task requires the model to determine the cellular location of a native protein. For instance, proteins inherently present in the lysosome will be designated with a category tag *"lysosome"*. Ten potential localizations exist, leading to the label $y \in \{0, 1, ..., 9\}$.

*Split:* We randomly split out a validation set from the training set with a 4:1 training/validation ratio.

**Binary Protein Localization (Bin).**

*Impact:* Identifying whether a protein is "soluble" or "membrane-bound" plays a pivotal role in comprehending its function. "Soluble" proteins operate as free molecules in organisms, while "membrane-bound" proteins might exhibit catalytic functions upon membrane attachment (Gimpelev et al., 2004). Efficiently distinguishing these two protein categories via computational methods can streamline biological research.

*Target:* The primary objective of the model in this task is a coarse classification of proteins into one of two categories: "membrane-bound" or "soluble". Consequently, the label for these proteins is $y \in \{0, 1\}$.

*Split:* For validation, we allocate a subset from the training data, maintaining a 4:1 ratio from training to validation. This task also assesses the model's capability to generalize over related protein sequences.

**Fluorescence Prediction (FLu).**

*Impact:* The green fluorescent protein acts as a crucial marker, allowing researchers to identify the existence of specific proteins in organic entities through its green glow (Tsien, 1998). This task aims to uncover the mutation trends that amplify or diminish such a biological characteristic.

*Target:* This challenge tasks the model to forecast the fitness of green fluorescent protein variants. The target label $y \in R$ corresponds to the logarithmic value of the fluorescence intensity as annotated by Sarkisyan et al. (Sarkisyan et al., 2016).

*Split:* We retain the division strategy from TAPE, emphasizing training on simpler mutants (with up to three mutations) and evaluating the model's performance on more complex mutants (with four or more mutations).

**Stability Prediction (Sta).**

*Impact:* The stability of a protein is pivotal for its functional efficacy in the body (Sarkisyan et al., 2016). This benchmarking task mirrors the practical application setting where functional mutants with satisfactory stability are chosen.

*Target:* The challenge here is to assess the stability of proteins in their natural environments. The target label $y \in R$ reflects the experimental stability measurement.

*Split:* We align with TAPE's splitting method, emphasizing training on proteins with multiple mutations and testing the model's capabilities on top-tier candidates having only a single mutation.

**$\beta$-lactamase activity prediction ($\beta$-lac).**

*Impact:* The TEM-1 beta-lactamase is the predominant enzyme granting gram-negative bacteria resistance to beta-lactam antibiotics (Palzkill & Botstein, 1992). This task delves into the improvement of this critical enzyme's activity through singular mutations.

*Target:* The aim is to analyze the activity among primary mutants of the TEM-1 beta-lactamase protein. The target label $y \in R$ corresponds to the empirically determined fitness score, which captures the proportional effect of mutations for each variant.

*Split:* High-capacity models are anticipated to discern proteins that differ by only a single amino acid residue in the dataset.

### Solubility prediction (Sol).

*Impact:* In the realms of pharmaceutical research and industry, protein solubility stands as a paramount attribute, as optimal solubility is indispensable for a protein's functionality (Khurana et al., 2018). This endeavor seeks to enhance the design of efficient computational tools that predict protein solubility based on sequences.

*Target:* The challenge revolves around forecasting a protein's solubility. To specify, it determines if a protein is soluble, resulting in a label $y \in \{0, 1\}$.

*Split:* The division of data evaluates the proficiency of models in extrapolating across diverse protein sequences.

### Contact prediction (Cont).

*Impact:* Estimating amino acid contacts derived from protein sequences is pivotal in predicting folded protein structures (Billings et al., 2021). This benchmark emphasizes medium- and long-range contacts, which play an instrumental role in the protein folding process.

*Target:* This assignment seeks to determine the contact likelihood between residue pairs. Each pair of residues is labeled with a binary value, $y \in \{0, 1\}$, signifying whether they establish contact within a predefined distance threshold $\delta$ or remain distant.

*Split:* In line with the CASP criteria (Kryshtafovych et al., 2019), our assessment hones in on the precision of the top L/5 contact predictions for medium and long-range contacts within the test dataset, thereby evaluating the prowess of contact prediction models in discerning the folded conformations of a diverse array of protein sequences.

### Fold classification (Fold).

*Impact:* Discerning the overarching structural topology of a protein at the fold level is invaluable for functional elucidation and drug design initiatives (Chen et al., 2016). Given that the SCOPe database (Fox et al., 2014) only categorizes a fractional segment of proteins in PDB (Berman et al., 2000a), there's a pronounced need to harness machine learning for automated fold classification directly from protein sequences.

*Target:* The objective is to classify the protein based on its global structural contour at the fold tier. This is denoted by a categorical label, $y \in \{0, 1, ..., 1194\}$, defined by the backbone coordinates of its structure.

*Split:* Superfamilies in entirety are excluded from the training phase and make up the test set. Such an arrangement offers a unique opportunity to assess the model's competency in recognizing proteins with structurally akin attributes but sequence differences, which is a hallmark of remote homology detection (Rao et al., 2019).

### Secondary structure prediction (SSP).

*Impact:* Accurately discerning the local structures of protein residues in their native conformation has multifaceted benefits, including insights into protein functionality (Klausen et al., 2019) and refining multiple sequence alignments (Simossis & Heringa, 2004). This benchmark exercise seeks to foster the development and testing of machine learning models tailored for such predictions.

*Target:* The mission is to prognosticate the local configurations of protein residues as they exist naturally. Each residue is earmarked with a secondary structure label $y \in \{0, 1, 2\}$, corresponding to coil, strand, or helix.

*Split:* While the primary source of data is Klausen's dataset for training, evaluation pivots on the CB513 dataset, ensuring a rigorous assessment of model generalization across variegated protein sequences.

**Yeast PPI prediction (Yst).**

*Impact:* Constructing comprehensive and accurate yeast interactome network maps is of paramount scientific significance (Yu et al., 2008; Pu et al., 2009; Baryshnikova et al., 2010). By forecasting binary yeast protein interactions using machine learning models, this benchmark task makes strides towards realizing this ambitious objective.

*Target:* The challenge mandates the model to ascertain whether a pair of yeast proteins engages in interaction. Pairs of proteins are designated with a binary label, symbolized as $y \in \{0, 1\}$, indicating the presence or absence of an interaction.

*Split:* We commence by pruning redundancies from all protein sequences in the dataset, setting a $90\%$ sequence identity threshold. Following this, these refined sequences are indiscriminately apportioned into training, validation, and test segments. Subsequently, redundancy elimination is carried out between each pair of these segments, with a stricter $40\%$ sequence identity cut-off. This ensures rigorous appraisal of the model's capacity for generalization across disparate protein sequences.

**Human PPI prediction (Hum).**

*Impact:* Deciphering the intricate web of the human protein interactome plays a crucial role in shedding light on disease mechanisms and unearthing novel disease-associated genes (Rual et al., 2005; Yu et al., 2011; Rolland et al., 2014). With this benchmark task, there's a hopeful anticipation of enhancing potent machine learning models adept at predicting human protein-protein interactions.

*Target:* The objective at hand is for the model to discern if a pair of human proteins is interactive. Each pairing is accompanied by a binary label, represented as $y \in \{0, 1\}$, indicating their interactive status.

*Split:* Our data partitioning strategy mirrors that of the yeast PPI prediction. However, we opt for an 8:1:1 division ratio for train, validation, and test segments, respectively. Just as before, this task evaluates the model's proficiency in generalizing across diverse protein sequences.

**PPI affinity prediction (Aff).**

*Impact:* The capability to forecast the relative binding vigor among potential binding candidates holds paramount importance in the realm of protein binder design (Liu et al., 2021; Shan et al., 2022). This task seeks to provide a pragmatic arena for machine learning models to demonstrate their efficacy in such a tangible application.

*Target:* The primary objective for the model is to compute the binding affinity, denoted as $y \in R$, gauged through pKd, between two protein entities.

*Split:* Delving deeper into the dataset segmentation, our training set amalgamates wild-type complexes alongside mutants possessing a maximum of two mutations. The validation set envelops mutants with a mutation count of three or four. Lastly, the test set encompasses mutants that exhibit more than four mutations. With this delineation, the task is positioned to assess the model's generalization prowess in a phased protein binder design context.

**PLI prediction on PDBbind (PDB).**

*Impact:* The elucidation of interactions between minor molecular entities and their corresponding target proteins emerges as a salient focus in drug discovery research (Yamanishi et al., 2010; Wen et al., 2017). This benchmark task is meticulously crafted to gauge the prowess of machine learning models in realizing this intricate objective.

*Target:* The onus is on the model to predict the interactions between small molecules and target proteins.

*Split:* To initiate, we diligently eradicate training sequences that parallel test sequences, deploying a 90% sequence identity threshold. Subsequently, the remaining training sequences undergo clustering. These clusters are then randomly apportioned into training and validation sets, abiding by a 9:1 distribution ratio. For assessment of model generalizability, the CASF-2016 benchmark (Su et al., 2018) is the chosen paradigm.

**PLI prediction on BindingDB (BDB).**

*Impact:* Recognizing the interactions between ligands and specific protein classes remains a pivotal endeavor in the realm of drug discovery. This benchmark task resonates with the aspirations of the drug discovery fraternity, emphasizing the evaluation of ligand interactions across four distinct protein classes.

*Target:* The core objective is for the model to ascertain ligand interactions, particularly focusing on four protein classes: ER, GPCR, ion channels, and receptor tyrosine kinases.

*Split:* The dataset segregation strategy, mirroring that of DeepAffinity, ensures that the aforementioned four protein classes are excluded from the training and validation phases, earmarking them exclusively for the generalization test.

## A.2 ATOM3D BENCHMARK

**Ligand Efficacy Prediction (LEP).**

*Impact:* Proteins often activate or deactivate by altering their form. Determining the shape a medication will encourage is pivotal in drug creation.

*Target:* This is approached as a binary classification challenge, where the goal is to ascertain if a molecule, when bound to these structures, will stimulate the protein's function.

*Split:* We categorize the complex pairs based on their protein targets.

**Small Molecule Properties (SMP).**

*Impact:* Estimating the physicochemical attributes of tiny molecules is a standard procedure in pharmaceutical chemistry and materials creation. While quantum-chemical assessments can reveal specific physicochemical characteristics, they are resource-intensive.

*Target:* Our goal is to forecast the properties of the molecules based on their ground-state configurations.

*Split:* We divide the molecules arbitrarily.

**Protein Structure Ranking (PSR).**

*Impact:* Proteins serve as fundamental agents within cells, and discerning their structure is typically vital for comprehending and tailoring their role.

*Target:* We approach this as a regression challenge, aiming to predict the global distance test for each structural blueprint relative to its experimentally defined structure.

*Split:* We segregate structures based on the year of competition.

**RNA Structure Ranking (RSR).** *Impact:* RNA, much like proteins, has pivotal functional responsibilities such as gene regulation and can take on distinct 3D configurations. However, the available data is limited, with only a handful of identified structures.

*Target:* Our aim is to estimate the root-mean-squared deviation (RMSD) for each structural model in comparison to its lab-verified structure.

*Split:* Structures are divided based on the respective year of the competition.

**Protein Interface Prediction (PIP).**

*Impact:* In many situations, proteins interact with one another. For instance, antibodies detect diseases by attaching to antigens. One fundamental challenge in comprehending these interactions is pinpointing the specific amino acids in two proteins that will engage upon binding.

*Target:* Our goal is to determine if two amino acids will come into contact when their parent proteins bind together.

*Split:* Protein complexes are divided ensuring that no protein from the training set shares above 30% sequence similarity with any protein in the DIPS validation set or the DB5 set.

**Ligand Binding Affinity (LBA).**

*Impact:* Proteins often modulate their functions by altering their structures. Determining the favored shape of a drug is crucial in the realm of drug design.

*Target:* We approach this as a binary classification challenge, aiming to discern if a molecule, when attached to these structures, will stimulate the protein's function.

*Split:* We categorize the complex pairs based on their specific protein targets.

**Residue Identity (RES).**

*Impact:* Comprehending the structural contribution of specific amino acids is pivotal for the creation of new proteins. This understanding can be achieved by forecasting the likelihood of various amino acids at a particular protein location, considering the adjacent structural backdrop (Torng & Altman, 2017).

*Target:* We approach this as a classification challenge, aiming to determine the central amino acid's identity by analyzing the surrounding atoms.

*Split:* We segregate environments based on the protein's topological category, as outlined in CATH 4.2 (Dawson et al., 2017), ensuring that environments from proteins of a similar class belong to the same divided dataset.

**Mutation Stability Prediction (MSP).**

*Impact:* Pinpointing mutations that reinforce protein interactions is crucial for crafting new proteins. Given that experimental methods to investigate these mutations are resource-intensive (Antikainen & Martin, 2005), there's a compelling need for streamlined computational approaches.

*Target:* We treat this as a binary classification challenge, aiming to determine if the complex's stability is enhanced due to the mutation.

*Split:* We categorize protein complexes ensuring that no protein in the evaluation set shares more than 30% sequence similarity with any protein in the instructional dataset.

## A.3  MULTI-LEVEL BENCHMARK

**Gene Ontology Term Prediction (GO).**

*Impact:* Accurately predicting a protein's functions using Gene Ontology (GO) terms is pivotal for enhancing our understanding of biological systems. By categorizing proteins based on their specific functions, we can gain deeper insights into cellular processes and mechanisms.

*Target:* The challenge lies in predicting multiple GO terms associated with a protein, effectively making it a multi-label classification task. Specifically, we delve into three ontologies: biological process (BP) with 1,943 categories, molecular function (MF) boasting 489 categories, and cellular component (CC) encompassing 320 categories.

*Split:* The dataset's division earmarks 29,898 proteins for training, 3,322 for validation, and 3,415 for testing. The F max accuracy metric is harnessed to evaluate the predictions.

**Enzyme Commission Number Prediction (EC).**

*Impact:* Predicting the Enzyme Commission (EC) numbers efficiently and accurately plays a pivotal role in understanding enzyme functions and categorizations. This task deviates from merely classifying enzyme reactions and strives to pinpoint the specific three-level and four-level 538 EC numbers, adding granularity to the enzymatic categorization.

*Target:* The overarching goal of this task is to predict the detailed EC numbers associated with each enzyme, which is a multi-label classification task. Such predictions give valuable insights into the specific reactions and pathways these enzymes partake in.

*Split:* The dataset splits are aligned with those detailed in (Gligorijević et al., 2021). As an additional note, in tasks like GO term and EC number predictions, measures are taken to ensure that the test set is comprised only of PDB chains with a sequence identity that doesn't exceed 95% compared to the training set, a standard adhered to in several studies such as (Wang et al., 2021b).

# B   DETAILS ON METRICS

**Pearson correlation.**   Pearson correlation (Pearson, 1900) is a statistic that measures the linear relationship between two continuous variables.

The mathematical formulation of Pearson correlation is given by:

$$R_P = \frac{\sum (X_i - \bar{X})(Y_i - \bar{Y})}{\sqrt{\sum (X_i - \bar{X})^2 \sum (Y_i - \bar{Y})^2}} \tag{6}$$

where $R_P$ is the Pearson correlation coefficient, $X_i$ and $Y_i$ are individual data points from the $X$ and $Y$ variables, $\bar{X}$ and $\bar{Y}$ are the averages of the $X$ and $Y$ variables.

**Spearman correlation.**   Spearman correlation (Spearman, 1904) is a statistical measure of the strength and direction of association between two variables. It is a non-parametric method used to assess the monotonic relationship between variables, meaning it doesn't assume a linear relationship between the variables as the Pearson correlation does.

The mathematical formulation of Spearman correlation is given by:

$$R_S = 1 - \frac{6 \sum d^2}{n(n^2 - 1)} \tag{7}$$

where $R_S$ is the Spearman correlation coefficient, $d$ is the difference between the ranks of each pair of corresponding values, $n$ represents the number of data points.

**Root Mean Square Error.**   Root Mean Square Error (RMSE) is a common metric used to evaluate the accuracy of a predictive model. It measures the average magnitude of the errors in a set of data.

RMSE is mathematically represented as:

$$RMSE = \sqrt{\frac{1}{n} \sum_{i=1}^{n} (y_i - \hat{y}_i)^2} \tag{8}$$

where $y_i$ is the actual or observed value for data point $i$, $\hat{y}_i$ is is the predicted value for data point $i$

**AUROC.**   Area Under the Receiver Operating Characteristic curve (AUROC) is a metric used to evaluate the performance of binary classification models. The ROC curve is a graphical representation of a model's ability to discriminate between the positive and negative classes across different threshold values. A higher AUROC suggests better model performance in distinguishing between the two classes.

**Protein-centric maximum F-score.**   Protein-centric maximum F-score ($F_{max}$) is the prediction probability for the $j$-th class of the $i$-th protein, $b_i^j \in \{0, 1\}$ is the corresponding binary class label and $J$ is the number of classes. F-Score is based on the precision and recall of the predictions for each protein.

$$\text{precision}_i(\lambda) = \frac{\sum_j^J ((p_i^J) \cap b_i^j)}{\sum_j^J (p_i^J \geq \lambda)}, \quad \text{recall}_i(\lambda) = \frac{\sum_j^J (p_i^J \geq \lambda)}{\sum_j^J b_i^j} \tag{9}$$

$F_{max}$ mathematically represented as:

$$\mathrm{F}_{max} = \max_{x \in [0,1]} \left\{ \frac{2 \times \text{precision}(\lambda) \times \text{recall}(\lambda)}{\text{precision}(\lambda) + \text{recall}(\lambda)} \right\} \tag{10}$$

where $\text{precision}(\lambda)$ and $\text{recall}(\lambda)$ represent average precision and recall over all proteins. They are defined as follows:

$$\text{precision}(\lambda) = \frac{\sum_i^N \text{precision}_i(\lambda)}{\sum_i^N ((\sum_j^J (p_i^J \geq \lambda)) \geq 1)}, \quad \text{recall}(\lambda) = \frac{\sum_i^N \text{recall}_i(\lambda)}{N} \tag{11}$$

