# OpenReview forum: "ProteinAdapter: Adapting Pre-trained Large Protein Models for Efficient Protein Representation Learning"
_ICLR.cc/2024/Conference — ICLR 2024 Conference Withdrawn Submission_

### Official Review · Reviewer_ZPKb · 2023-10-28

**Soundness:** 1 poor
**Presentation:** 1 poor
**Contribution:** 2 fair
**Rating:** 3
**Confidence:** 3

**Summary:**

Numerous recent works have proposed pre-trained large protein models based on either protein sequences or protein structures. This work proposes a parameter-efficient fine-tuning method called ProteinAdapter. Based on cross-attention blocks, ProteinAdapter merges representations from multiple levels, i.e., ESM1b representations based on 1D sequence and ESM-IF1 representations based on 3D structures. Additionally, the authors propose a multi-scale predictor on top of the ProteinAdapter to extract condensed final representations hierarchically. They claim that the proposed method outperforms state-of-the-art methods in over 20 tasks under both single-task and multi-task settings.

**Strengths:**

The authors proposed an interesting approach to merge representations of two protein language models based on sequences and structures. While this is not the first hybrid approach to protein language models (ESM-GearNet proposed sequential pre-training of sequence and structures), it seems novel. If the proposed method is effective in terms of efficiency and performance, as the authors claimed, it will benefit a broad range of researchers working on various protein biology tasks.

**Weaknesses:**

- [Novelty] Utilizing cross-attention for multi-modal representations is a fairly common approach, particularly for vision-language models. It seems the paper did not discuss the relevant previous works properly.
- [Introduction] The paper generally describes the ProteinAdapter as it takes the representations from ESM-1b and ESM-IF1 (e.g. in Intro). However, since a significant amount of experiments conducted on PEER do not utilize ESM-IF1, I think the paper is quite misleading.
- [Related Work] I believe stating ESM-GearNet as non-parameter-efficient fine-tuning method is not appropriate. The focus of the work was more on the subsequent pretraining on protein structures from the AlphaFold Database, rather than fine-tuning for downstream tasks.
- [Methods] The paper lacks explanations on ESM-1b and ESM-IF1 even though they are critical components throughout the manuscript.
- [Training Setup] I’m confused about which method the authors used to cope with the lack of structural information for the PEER benchmark. Can you confirm that the authors simply replaced the structural embedding with another layer of sequence embedding throughout the paper?
- [Training Setup] What do you mean by “we only conducted multi-level performance tests on the Stability Prediction and Fold Classification tasks?” What are the results in Table 2-4? It seems quite misleading.
- [Sec 4.3 Multi-Task setting] How do you define which one is primary among multi-tasks? It seems the authors did not try to decrease the loss from supportive tasks by setting alpha less than 1. In my view, since the authors stated "multi-task", the model must be evaluated on multiple tasks rather than a single primary task.
- [Table 2-3] It's unclear how much the proposed ProteinAdapter contributes to the performance improvement. For Table 2, can you provide a comparison with other ways to use representations from multiple layers? For Table 3, can you provide a comparison with the fine-tuning performance of ESM-IF1?
- [Table 3] I thought the authors used embeddings from a different layer of ESM1b instead of structural embeddings. If so, interpreting results as “It can be observed that since
our method has already made full use of the structural information, using structure prediction tasks as an auxiliary does not significantly improve the performance of the central task. However,” does not seem like a logical explanation.
- [Table 5] It does not seem like a fair comparison. Simply removing the adapter or predictor significantly decreases model capability (in terms of parameters). There should be more competitive baselines instead of off-the-shelf removals. For example, for the w/o multi-scale predictor, how about just removing the final merge operation?
- [Baselines] How did the authors conduct baseline experiments in Table 2-5? If they, fine-tuned the entire pretrained model ESM-1b on downstream tasks, it would not be a common practice. Can you provide more baseline results by fine-tuning 1-2 dense layers on top of the frozen pretrained model?

Minor comments
- [Missing references] (1) In the Introduction section, references are missing for ESM-1b and ESM-IF1 in the IN. The references are provided later on. (2) In the Methods section, it seems references are needed for the statement “due to genetic differences, the scale of the same functional regions is often inconsistent across different species.”

**Questions:**

- [Methods] What's the rationale behind using ESM-IF1 representations as the structural branch, considering it is trained for inverse folding rather than general pre-trained representations?
- [Baselines] For Table 2, can you provide a comparison with other ways to use representations from multiple layers? For Table 3, can you provide a comparison with the fine-tuning performance of ESM-IF1?
- [Table 5] It does not seem like a fair comparison. Can the authors provide more competitive baselines instead of off-the-shelf removals?
- [Baselines] How did the authors conduct baseline experiments in Table 2-5? Can you provide more baseline results by fine-tuning 1-2 dense layers on top of the frozen pretrained model?

---

### Official Review · Reviewer_PNTU · 2023-10-29

**Soundness:** 3 good
**Presentation:** 3 good
**Contribution:** 2 fair
**Rating:** 3
**Confidence:** 3

**Summary:**

This paper ensemble current protein pretraining models to enhance multiple downstream tasks. The ESM-1b and ESM-IF are utilized as sequence and structure pretraining models, respectively. Based on the pretraining embeddings, the authors develop an adapter model equipped with a multi-scale predictor to support over 20 downstream tasks.

**Strengths:**

The paper is easy to follow and the presentation is clear. The authors provide extensive experiments to confirm the effectiveness of the proposed methods.

**Weaknesses:**

1. The novelty is limited. Integrating pretraining models to enhance downstream tasks is a trivial idea. The main contribution comes from engineering labor.
2. The relative improvements to SOTA methods are not significant on the PEER benchmark.  In Table 2, the proposed methods could not guarantee better performance than previous SOTAs. Sometimes, the achieved results are even worse than SOTAs.

**Questions:**

1. Could the authors compare ProteinAdapter to ESM-GearNet [1]?
[1] A Systematic Study of Joint Representation Learning on Protein Sequences and Structures

2. Could the authors provide ablation about Intra-Level Self-Attention? What will happen if you use separate embedding layers to project sequence and structure features and then add them together to feed into downstream transformers?

---

### Official Review · Reviewer_XhEm · 2023-10-31

**Soundness:** 3 good
**Presentation:** 3 good
**Contribution:** 2 fair
**Rating:** 5
**Confidence:** 4

**Summary:**

This paper proposes ProteinAdapter for efficient protein representation learning by adapting pre-trained large protein models. The method leverages a multi-level adapter to integrate sequence and structure embeddings from models like ESM-1b and ESM-IF1. It also uses a multi-scale predictor to handle proteins at different scales. Experiments on over 20 tasks show performance gains over previous methods.

**Strengths:**

1. The idea of efficiently adapting large pre-trained protein models is practically useful and intuitive.
2. The method achieves new state-of-the-art results on major protein analysis benchmarks like PEER and ATOM3D.
3. The adapter design leads to computational efficiency with much fewer parameters compared to training large models from scratch.
4. This paper is well-written and structured. The motivation and technical details are clearly explained.

**Weaknesses:**

1. The efficiency gains of the adapter are not quantitatively analyzed. Concrete speedup measurements could be provided.
2. More analysis could be provided on how the adapter exactly helps integrate and preserve information across levels.
3. The novelty is somewhat limited. I do believe such a work is meaningful for computational biology, but the significance in AI or ML community is limited.

**Questions:**

1. How are very long protein sequences handled with the pyramid architecture?
2. Is this adapter architecture specific to protein models or could it generalize to other domains?

---

### Official Review · Reviewer_vd8N · 2023-11-01

**Soundness:** 2 fair
**Presentation:** 3 good
**Contribution:** 3 good
**Rating:** 5
**Confidence:** 5

**Summary:**

This paper focuses on the task of protein representation learning and aims to adapt pre-trained large protein models (LPM) to this task.

Specifically, the authors want to do parameter-efficient fine-tuning, only training a few added trainable parameters.

In terms of the model architecture, they use two (fixed/freezed) LPMs (ESM-1b for protein sequences and ESM-IF1 for protein 3D structures) as the base model and add two more (trainable) blocks.

The first block is Multi-Level ProteinAdapter which combines the sequence and structure representations with self- and cross-attention layers.
The second block is Multi-Scale Predictor which uses several pooling layers to learn different-scale representations.

The model is fine-tuned on various downstream tasks and can achieve good results.

**Strengths:**

1. This paper is well written and easy to follow. The illustration figures are clear.
2. The results look good, overcoming previous methods on most tasks.

**Weaknesses:**

1. About the datasets

I feel that the used datasets are not suitable for the proposed method. One main contribution of the method is the ProteinAdapter which combines the sequence and structure representations, therefore, I think the authors should conduct experiments on protein datasets with both sequence and structure information as input. However, the used datasets don’t contain such information.
 - PEER benchmark is designed for protein sequence understanding and only provides sequence information.
 - ATOM3D provides 3D coordinates for each atom, but it includes small molecules, proteins, RNA, etc, not only proteins. In addition, I am wondering how the proposed method can be applied to ATOM3D datasets. Do the authors directly take each atom as a node?

2. About the baseline methods and results

More protein-specified tasks and baseline methods (IEConv[1], GVP[2][3], GearNet[4], ProNet[5], CDConv[6], etc) should be included in the experiment part.

3. Comparison to other methods:

IEConv and CDConv also use pooling layers to reduce the number of nodes and learn different scale representations. ProNet provides hierarchical representations for proteins. What is the difference between the proposed multi-scale hierarchical predictor and these previous methods?

[1] Intrinsic-Extrinsic Convolution and Pooling for Learning on 3D Protein Structures
[2] Learning from Protein Structure with Geometric Vector Perceptrons.
[3] Equivariant Graph Neural Networks for 3D Macromolecular Structure.
[4] Protein Representation Learning by Geometric Structure Pretraining.
[5] Learning Hierarchical Protein Representations via Complete 3D Graph Networks.
[6] Continuous-Discrete Convolution for Geometry-Sequence Modeling in Proteins

**Questions:**

See the weaknesses part.

My main concern is the used dataset. I think they are really not suitable for the proposed method.